# High Risk of Peritonsillar Abscess in End-Stage Renal Disease Patients: A Nationwide Real-World Cohort Study

**DOI:** 10.3390/ijerph18136775

**Published:** 2021-06-24

**Authors:** Geng-He Chang, Ang Lu, Yao-Hsu Yang, Chia-Yen Liu, Pey-Jium Chang, Chuan-Pin Lee, Yao-Te Tsai, Cheng-Ming Hsu, Ching-Yuan Wu, Wei-Tai Shih, Ming-Shao Tsai

**Affiliations:** 1Department of Otolaryngology—Head and Neck Surgery, Chiayi Chang Gung Memorial Hospital, Chiayi 613, Taiwan; genghechang@gmail.com (G.-H.C.); satan54213@gmail.com (A.L.); yaote1215@gmail.com (Y.-T.T.); scm00031@gmail.com (C.-M.H.); 2Health Information and Epidemiology Laboratory, Chiayi Chang Gung Memorial Hospital, Chiayi 613, Taiwan; r95841012@ntu.edu.tw (Y.-H.Y.); qchiayen@gmail.com (C.-Y.L.); cblee@cgmh.org.tw (C.-P.L.); 3Graduate Institute of Clinical Medical Sciences, College of Medicine, Chang Gung University, Taoyuan 333, Taiwan; peyjiumc@mail.cgu.edu.tw; 4Department of Traditional Chinese Medicine, Chiayi Branch of Chang-Gung Memorial Hospital, Chiayi 613, Taiwan; smbepigwu77@gmail.com (C.-Y.W.); ati8955@cgmh.org.tw (W.-T.S.); 5School of Traditional Chinese Medicine, College of Medicine, Chang Gung University, Taoyuan 333, Taiwan; 6Faculty of Medicine, College of Medicine, Chang Gung University, Taoyuan 333, Taiwan

**Keywords:** tonsillitis, cellulitis, kidney disease, predisposing factor, risk factor

## Abstract

Background: Peritonsillar abscess (PTA) is an infectious emergency in the head and neck, and patients with end-stage renal disease (ESRD) have an immunocompromised status. However, no relevant research has focused on the ESRD–PTA relationship. This study explored PTA in ESRD patients and their prognosis. Methods: We identified 157,026 patients diagnosed as having ESRD over January 1997 to December 2013 from Taiwan’s National Health Insurance Research Database (NHIRD). Each patient with ESRD (hereafter, patients) was matched with one control without chronic kidney disease (CKD; hereafter, controls) by sex, age, urbanization level, and income. Next, PTA incidence until death or the end of 2013 was compared between the two groups, and the relative risk of PTA was analyzed using a multiple logistic regression model. Results: The patients had a significantly higher PTA incidence than did the controls (incidence rate ratio: 2.02, 95% confidence interval [CI]: 1.40–2.91, *p* < 0.001). The Kaplan–Meier analysis revealed that the patients had a higher cumulative incidence of PTA than did the controls (*p* < 0.001). In Cox regression analysis, the patients had nearly twofold higher PTA risk (adjusted hazard ratio [HR]: 1.98, 95% CI: 1.37–2.86, *p* < 0.001). The between-group differences in the PTA-related hospital stay length (8.1 ± 10.3 days in patients and 5.7 ± 4.6 days in controls, *p* = 0.09), consequent deep-neck infection complication (4.2% in patients and 6.3% in controls, *p* = 0.682), and mortality (0.0% in both groups) were nonsignificant. **Conclusions:** Although ESRD does not predict a poor prognosis of PTA, it is an independent PTA risk factor.

## 1. Introduction

Peritonsillar abscess (PTA) is a head and neck infection caused by pus accumulation in the space between the lateral side of the palatine tonsils and the superior pharyngeal constrictor muscles [1]. Patients with PTA typically present to the emergency department with fever, severe sore throat, odynophagia, dysphagia, trismus, and even, respiratory distress, and they often require pus drainage, antibiotic therapy, and supportive therapy [2,3].

Deep neck infection (DNI) is caused by bacteria invading the deep neck space and is a fatal emergency. PTA is different from DNI in terms of clinical manifestations, infection source, and therapy. For instance, incision and drainage plus antibiotic therapy, tonsillectomy à chaud, or interval tonsillectomy are recommended treatment for PTA, whereas intensive medical care, intravenous antibiotics, and aggressive therapies (e.g., surgical debridement or even tracheostomy) are indicated for DNIs [4,5,6]. In addition, PTA typically occurs due to recurrent tonsillitis, whereas DNIs are mostly caused by odontogenic infection [7,8,9,10,11]. Severe PTA can progress to the parapharyngeal space and lead to a DNI [12].

In patients with end-stage renal disease (ESRD), immunodeficiency prevalence, infection risk, mortality, and morbidity are higher than in those without chronic kidney disease (CKD). We previously established that ESRD is a potential predisposing factor for DNI [13] and can increase a higher mortality rate in affected patients compared with that in patients without CKD. However, few studies have investigated the correlation between ESRD and PTA. Therefore, the association of ESRD with PTA occurrence and prognosis was explored in this study.

## 2. Methods

### 2.1. Source of Study Data

National Health Insurance (NHI) Research Database (NHIRD), established by the Taiwan government, contains the claims data of NHI beneficiaries, who represent approximately 99% of Taiwan’s population [14]. The claims data included disease diagnoses received during each outpatient visit and hospitalization, drug prescriptions and dosages, examination items, and operation methods, and related medical expenses. It also contains data (generated and saved as electronic files) on the beneficiaries’ income level, residence, and other details [15]. Therefore, NHIRD is a population-level database used to generate real-world evidence to support clinical decision-making and health care policymaking [16]. In NHIRD, the diagnoses are coded using the International Classification of Diseases, Ninth Revision, Clinical Modification codes (ICD-9-CM) [13]. Moreover, the NHIRD claims data are deidentified to ensure that the beneficiaries’ privacy is ensured. Chang Gung Memorial Hospital’s Institutional Review Board approved this study (approval number: 201901365B0).

### 2.2. Study Cohort

The NHI system lists ESRD as a catastrophic illness, and patients with chronic renal failure who need regular hemodialysis are registered as Catastrophic Illness Patients (CIP) [13,17] to reduce their related medical expenses considerably. However, to receive this certification, the process involves rigorous and careful medical evaluation, along with the submission of all relevant documents to the pertinent authorities for corroboration. Moreover, only patients with chronic renal failure who require regular dialysis for at least 3 months are eligible for certification [13]. The data from the Registry of CIP (RFCIP), part of NHIRD, can thus be used to reliably identify ESRD patients with CIP certification. Hence, we enrolled all RFCIP-registered patients diagnosed as having ESRD (ICD-9-CM: 585, 586, 403.01, 403.11, 403.91, 404.02. 404.03, 404.12, 404.13, 404.92, and 404.93) between 1997 and 2013 (Figure 1) [13].

### 2.3. Comparison Cohort

The Longitudinal Health Insurance Database 2005 (LHID2005), an NHIRD subset, contains the data of 1,000,000 beneficiaries randomly chosen from all NHI beneficiaries in Taiwan in 2005—according to the report of the National Health Research Institutes. Some validation studies have pointed out the absence of significant differences in the sex and age distribution between the LHID2005 population and all NHI beneficiaries [15,18,19,20]. Thus, LHID2005 has been used by many studies for representing the national population. Finally, to form a non-CKD cohort for comparison without kidney disease, we enrolled individuals without CKD from LHID2005.

### 2.4. Matching Process

To creating the ESRD and non-CKD group, each patient with ESRD was matched an individual without CKD from LHID2005 by age, sex, income, residence urbanization level, and presence of diabetes mellitus (DM) and hypertension. For every patient with ESRD, the index date was the date of registration in RFCIP due to ESRD, and the same index date was used for their matched non-CKD control; this provided a reference to assess the development of primary study outcome.

### 2.5. PTA Incidence as Main Study Outcome

PTA occurrence was the main outcome, the occurrence of which was defined by the main diagnosis of PTA (ICD-9-CM: 475) received at hospitalization [21]. All enrollees were followed from the index date until the date of PTA occurrence or death or 31 December 2013.

### 2.6. Comorbidities

The following comorbidities were considered for all enrollees: chronic tonsillitis (ICD-9-CM: 474.00–474.02), DM (ICD-9-CM: 250), hypertension (ICD-9-CM: 401–405), chronic obstructive pulmonary disease (COPD; ICD-9-CM: 491,492, 494, and 496), liver cirrhosis (LC; ICD-9-CM: 571.2 and 571.5–571.6), systemic autoimmune diseases (ICD-9-CM: 696.0, 696.1, 710.0, 710.1, 710.2, and 710.4), and dyslipidemia (ICD-9-CM: 272) [11,13,14,17,19,22]. We included a comorbidity in the analysis if the enrollee had more than one relevant inpatient diagnosis or more than three relevant outpatient diagnoses in the past.

### 2.7. Treatment Modalities and Prognosis Evaluation

The treatment modalities including antibiotics, transoral needle aspiration/incision and drainage, and tonsillectomy were analyzed. The prognosis was assessed by analyzing between-group differences in hospital stay length and occurrence of DNI and mediastinitis. Here, DNI and mediastinitis occurrence was defined according to hospitalizations for DNIs: parapharyngeal abscess (ICD-9-CM: 478.22), retropharyngeal abscess (ICD-9-CM: 478.24), Ludwig angina (ICD-9-CM: 528.3), or cellulitis and abscess of the neck (ICD-9-CM: 682.1), and mediastinitis (ICD-9-CM: 519.2) [11,13,17,23].

### 2.8. Statistical Analysis

To compare comorbidities and demographic characteristics between the ESRD and non-CKD groups, we used unpaired Student *t* and Pearson chi-square tests for continuous and categorical variables, receptively. Next, we employed Kaplan–Meier analysis to assess the cumulative incidence of PTA and determined the differences by using a two-tailed log test. We used a multivariate Cox regression model to analyze the risk ratio (hazard ratio [HR]) for PTA between the ESRD and non-CKD groups. In addition, we conducted sensitivity and subgroup analyses to detect whether ESRD and comorbidities interact to inhibit or promote PTA development. For all analyses, SAS (version 9.4; SAS Institute, Cary, NC, USA) was used and a *p* of <0.05 was considered to indicate statistical significance.

## 3. Results

As presented in Table 1, the ESRD and non-CKD groups included 103,141 and 103,141 individuals. After 1:1 matching by sex, age, socioeconomic status, and hypertension and DM, the ESRD group had a significantly higher prevalence of chronic tonsillitis, LC, and autoimmune diseases than did the non-CKD group. Moreover, PTA incidence was significantly higher in the ESRD group (*p* = 0.028). In the ESRD and non-CKD groups, the PTA incidence rate was 9.7 and 4.8 per 100,000 person-years, respectively (mean follow-up duration = 9.2 ± 4.7 and 9.7 ± 4.5 years, respectively; Table 2).

In the ESRD group, the overall incidence rate ratio (IRR) was 2.02 (95% confidence interval [CI]: 1.40–2.91, *p <* 0.001), mean duration from ESRD diagnosis to PTA development was 4.6 ± 3.5 years. When dividing the follow-up duration was divided into <1, 1–5, and ≥5 post-ESRD diagnosis years, the IRRs (95% CIs) were 2.48 (0.87–7.03, *p* = 0.088), 2.23 (1.24–4.01, *p* = 0.008), and 1.72 (1.01–2.93, *p* = 0.047), respectively.

The Kaplan–Meier analysis (Figure 2) showed a significantly higher cumulative incidence of PTA in the ESRD group than in the non-CKD group (*p* < 0.001, log-rank test). According to Cox proportional hazard model and after adjustment for gender, age, urbanization, and income level, PTA risk was approximately twofold in the ESRD group than in the non-CKD group (adjusted HR [95% CI] for the main model: 1.98 [1.37–2.86], *p* < 0.001; Table 3). Our sensitivity analyses involved alternate addition of each comorbidity to the main model; the results indicated a considerably constant PTA risk among ESRD patients. Moreover, subgroup analyses revealed that a relatively stable effect of ESRD on the PTA risk was in the subgroups.

The treatment modalities and prognosis of PTA in both groups were presented in Table 4. In the ESRD-PTA group and non-CKD-PTA group, 100% and 95.8% of the patients received antibiotic treatment, respectively. There was no significant difference in the proportion of the ESRD-PTA group and Non-CKD-PTA group receiving transoral needle aspiration/incision and drainage (37.5% vs. 52.1%, *p* = 0.682). Of the two groups, only one patient in the ESRD-PTA group underwent tonsillectomy. Regarding prognosis, hospitalization due to PTA was longer in the ESRD group than in the non-CKD group; however, this difference was nonsignificant (8.1 ± 10.3 vs. 5.7 ± 4.6 days, *p* = 0.09). In addition, the proportion of PTA complicated with DNI was similar between the ESRD and non-CKD groups (4.2% and 6.3%, respectively; *p* = 0.682). One patient in each of the ESRD and non-CKD groups progressed to mediastinitis (1.4% and 2.1%, respectively; *p* = 0.773). PTA-associated mortality was noted in neither group, and only one patient in the ESRD group experienced PTA recurrence within 3 months.

## 4. Discussion

This nationwide study was based on real-world evidence and is the first study to explore the effect of ESRD on PTA. Our study results indicated PTA risk was approximately twofold higher in patients with ESRD than in those without CKD. Moreover, PTA incidence increased nearly 2.5-fold within the first 5 years of ESRD diagnosis.

When encountering ESRD patients complaining of sore throat and difficulty swallowing, clinicians should pay special attention to peritonsillar swelling and uvula deviation, so that PTA can be diagnosed and treated as soon as possible. Patients with ESRD typically have infectious complications due to multifactorial mechanisms, such as neutrophil dysfunction, uremic toxicity, biological incompatibility of the dialyzer membrane, anemia, and iron overload [24,25]. Patients undergoing chronic dialysis have been reported to have higher risks of hemodialysis vascular access device infection, bacteremia, pneumonia, urinary tract infection, and peritonitis [26,27]. Moreover, our previous study demonstrated that ESRD is an independent risk factor for DNI and that approximately twofold higher risk of DNI in patients with ESRD than in controls [13]. Accordingly, patients with renal failure have a higher PTA risk than healthy people.

Under normal conditions, patients with PTA receive treatment during hospitalization, with the hospital stay length ranging from 2 to 7 days [3,28,29]. Patients with medical comorbidities or older patients have longer hospital stay lengths. In our study, ESRD patients with PTA had a longer hospital stay (8.1 ± 10.3 days) than that reported previously, but the PTA hospital stay length for non-CKD patients was within the normal range. Although the difference in the hospital stay length between the ESRS–PTA and non-CKD–PTA groups did not reach statistical significance, the results imply that patients with ESRD–PTA require a longer hospital stay for undergoing treatment.

The major complication of PTA is the abscess extending laterally to the parapharyngeal space to form DNI, but occasionally, the abscess might invade into the mediastinum and cause necrotizing mediastinitis, and it may even invade into the prevertebral space to result in cerebral abscess formation or meningitis development [30]. Therefore, once PTA progresses to DNI and mediastinitis, the mortality rate will increase substantially [31]. According to the literature, approximately 2% of PTA may be complicated by DNI, but few studies have explored whether comorbidities increase the chance of the PTA-DNI complication [32]. In our study, PTA was complicated with DNI in 4.2% of the hemodialysis patients—higher than the previously reported prevalence. However, because the number of people with DNI in our study was too small, meaningful statistical analysis could not be conducted, and further research is required for investigating this further.

The two groups received similar treatments, and most patients received antibiotics alone or transoral needle aspiration/incision and drainage combined with intravenous antibiotics. Although the results of this study showed that only 1% of PTA patients in Taiwan received tonsillectomy, there is now more and more evidence supporting that tonsillectomy is a safe and effective primary treatment for PTA [4,6]. The complications of PTA including DNI and mediastinitis showed no difference between both groups. There were no mortalities in both ESRD-PTA and non-CKD-PTA groups. The outcomes were favorable, and this might be related to the easy approach to the abscess in the peritonsillar space for drainage compared with the deep-neck space, and this attenuated the impact of ESRD on the PTA mortality.

Our study has several strengths. First is the large sample of ESRD patients and its comparison with a database representing the national population of Taiwan; this enabled real-world analysis of the evidence. Furthermore, with a long follow-up period, we could analyze the time from the diagnosis of ESRD to the occurrence of PTA, and we could divide the follow-up time to explore the incidence of PTA at different time points; the results revealed that the initial stage after diagnosis had the highest risk of PTA infection. However, our research also has some limitations. First, this real-world database does not contain information on clinical symptoms and physical examination outcomes. Second, detailed information such as medical records, imaging examinations, and laboratory data are not available; thus, we could not assess the severity and extent of the abscess. Third, we could not obtain bacterial culture results; thus, we were unable to discuss the type of bacteria that caused the infection and drug sensitivity. Finally, although Kim et al. reported smoking and drinking are risk factors for PTA, our study could not evaluate the effects of smoking and drinking on PTA because NHIRD did not provide this information [33]. A study including the missing information should be conducted to further investigate the research topic.

## 5. Conclusions

In this nationwide study, PTA risk was higher in patients with ESRD than in those without CKD; moreover, the risk of PTA was significantly higher within the first 5 years of ESRD onset. However, with appropriate treatment, the prognosis of ESRD patients with PTA was favorable. In conclusion, ESRD is an independent risk factor, but not a poor prognostic factor, for PTA. Clinicians should pay more attention to PTA risk when caring for patients with ESRD.

## Figures and Tables

**Figure 1 ijerph-18-06775-f001:**
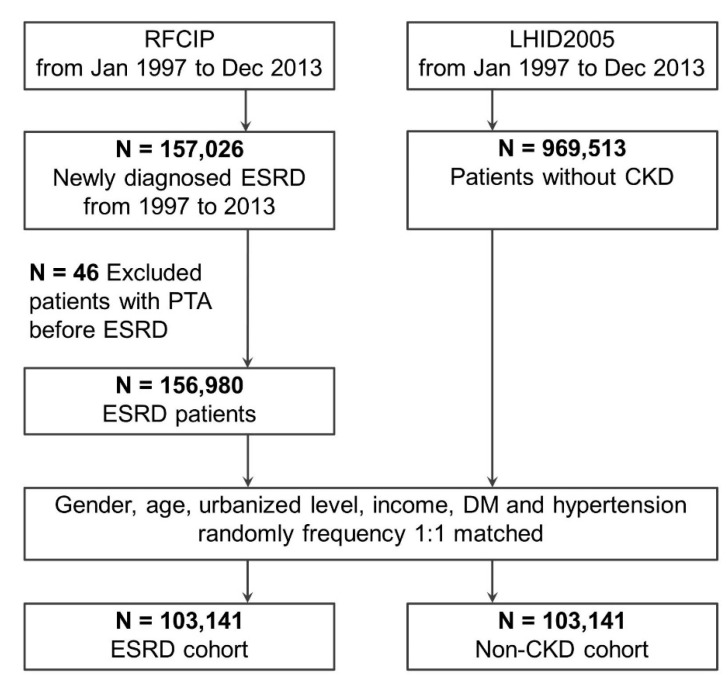
Flow of study and comparison cohort enrollment. Abbreviations: CKD, chronic kidney disease; DM, diabetes mellitus; ESRD, end-stage renal disease; LHID2005, Longitudinal Health Insurance Database 2005; RFCIP, Registry of Catastrophic Illness Patients; PTA, peritonsillar abscess.

**Figure 2 ijerph-18-06775-f002:**
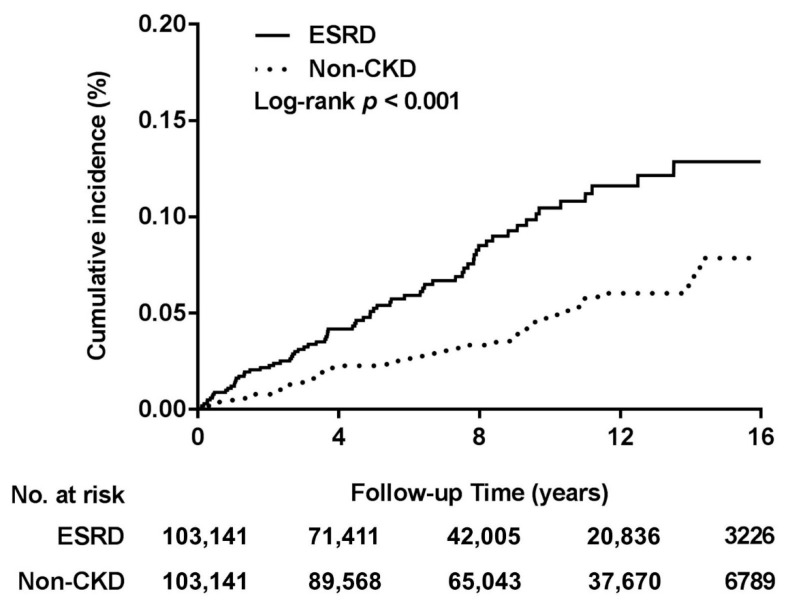
Cumulative incidence of PTA in the ESRD and non-CKD groups. Abbreviations: CKD, chronic kidney disease; ESRD, End-stage renal disease.

**Table 1 ijerph-18-06775-t001:** Demographic and clinical characteristics.

Characteristic	ESRD		Non-CKD		*p*-Value
	N	%	N	%	
Total	103,141		103,141		
Sex					1.000
Male	51,306	49.7	51,306	49.7	
Female	51,835	50.3	51,835	50.3	
Age (years)					1.000
<65	60,364	58.5	60,364	58.5	
≥65	42,777	41.5	42,777	41.5	
Urbanized level					1.000
1 (City)	25,742	25.0	25,742	25.0	
2	45,506	44.1	45,506	44.1	
3	19,407	18.8	19,407	18.8	
4 (Village)	12,486	12.1	12,486	12.1	
Income (NTD/month)					1.000
0	25,950	25.2	25,950	25.2	
1–15,840	21,737	21.1	21,737	21.1	
15,841–25,000	43,320	42.0	43,320	42.0	
≥25,001	12,134	11.8	12,134	11.8	
Comorbidities					
Chronic tonsillitis	88	0.1	213	0.2	<0.001
DM	51,292	49.7	51,292	49.7	1.000
Hypertension	93,712	90.9	93,712	90.9	1.000
COPD	24,631	23.9	24,701	24.0	0.718
LC	10,381	10.1	4244	4.1	<0.001
Autoimmune	4387	4.3	3232	3.1	<0.001
Dyslipidemia	48,067	46.6	49,688	48.2	<0.001
Outcome					
PTA	72	0.070	48	0.047	0.028

Abbreviations: CKD, chronic kidney disease; COPD, chronic obstructive pulmonary disease; DM, diabetes mellitus; ESRD, end-stage renal disease; LC, liver cirrhosis; NTD, New Taiwan Dollar; PTA, peritonsillar abscess.

**Table 2 ijerph-18-06775-t002:** PTA incidence.

	ESRD				Non-CKD				IRR (95% CI)	*p*-Value
	N	PTA	PYs	Rate †	N	PTA	PYs	Rate †		
Overall follow-up years	103,141	72	745,775	9.7	103,141	48	1002,588	4.8	2.02 (1.40–2.91)	<0.0001
<1	103,141	12	99,510	12.1	103,141	5	102,687	4.9	2.48 (0.87–7.03)	0.0884
1–5	95,513	32	317,802	10.1	101,912	17	376,309	4.5	2.23 (1.24–4.01)	0.0076
>5	63,109	28	328,463	8.5	83,810	26	523,593	5.0	1.72 (1.01–2.93)	0.0472

Abbreviations: CKD, chronic kidney disease; ESRD, end-stage renal disease; IRR, incidence rate ratio; PYs, person-years. † Rate: per 100,000 person-years; IRR was compared using Poisson regression.

**Table 3 ijerph-18-06775-t003:** Multivariable Cox proportional hazard regression for the association of ESRD with PTA and covariates.

Variables	Adjusted HR (95% CI)	*p*-Value
Multivariable Regression Analysis		
Non-CKD	Reference	
ESRD (main model *)	1.98 (1.37–2.86)	<0.001
Sensitivity analysis ^†^		
Main model + Chronic tonsillitis	1.99 (1.38–2.88)	<0.001
Main model + DM	2.01 (1.39–2.90)	<0.001
Main model + Hypertension	2.00 (1.39–2.89)	<0.001
Main model + COPD	1.99 (1.38–2.87)	<0.001
Main model + LC	2.01 (1.39–2.90)	<0.001
Main model + Autoimmune	2.01 (1.39–2.90)	<0.001
Main model + Dyslipidemia	1.98 (1.37–2.86)	<0.001
Subgroup analysis		
Sex		
Female	2.38 (1.28–4.24)	0.006
Male	1.79 (1.12–2.85)	0.015
Age		
<65	2.22 (1.41–3.50)	0.001
≥65	1.56 (0.83–2.95)	0.167
Chronic tonsillitis		
No	1.94 (1.35–2.81)	<0.001
Yes	–	–
DM		
No	2.31 (1.33–3.99)	0.003
Yes	1.79 (1.09–2.95)	0.022
Hypertension		
No	1.31 (0.50–3.42)	0.587
Yes	2.15 (1.44–3.21)	<0.001
COPD		
No	1.96 (1.30–2.95)	0.001
Yes	2.16 (0.92–5.05)	0.076
LC		
No	2.03 (1.39–2.97)	<0.001
Yes	1.39 (0.28–6.97)	0.687
Autoimmune		
No	2.06 (1.43–2.99)	<0.001
Yes	–	–
Dyslipidemia		
No	1.86 (1.13–3.04)	0.014
Yes	2.18 (1.25–3.78)	0.006

Abbreviations: CI, confidence interval; ESRD, end-stage renal disease; HR, hazard ratio; PTA, peritonsillar abscess. * Main model, adjusted for sex, age, urbanization, and income level. ^†^ Model adjusted for covariates in the main model and for each additional listed comorbidities.

**Table 4 ijerph-18-06775-t004:** The treatment modalities and prognosis of PTA in ESRD and non-CKD groups.

	ESRD-PTA		Non-CKD-PTA		
	N = 72		N = 48		
	N	%	N	%	*p*-Value
Treatment					
Antibiotic	72	100.0	46	95.8	- ^#^
Aspiration	27	37.5	25	52.1	0.135
Tonsillectomy	1	1.4	0	0.0	- ^#^
Prognosis					
Hospitalization (days)	8.1 ± 10.3		5.7 ± 4.6		0.090
Progress to DNI	3	4.2	3	6.3	0.682
Progress to mediastinitis	1	1.4	1	2.1	0.773
Mortality rate	0	0.0	0	0.0	- ^#^
Recurrence within 3 months	1	1.4	0	0.0	- ^#^

Abbreviations: DNI, deep neck infection; ESRD, end-stage renal disease; PTA, peritonsillar abscess. ^#^ The *p*-value cannot be calculated when the incidence rate of one group is 0 or 100%.

## Data Availability

The datasets analyzed in the current study are available in the Taiwan National Health Insurance Research Database repository (https://nhird.nhri.org.tw/en/).

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
