# Peer review of "High Risk of Peritonsillar Abscess in End-Stage Renal Disease Patients: A Nationwide Real-World Cohort Study"

_ijerph, 2021, doi:10.3390/ijerph18136775_

Round 1

Reviewer 1 Report

Please  correction: 

In the introduction section must clearly state and emphasize that, according to current recommendations, the treatment of PTA is tonsillectomia  a chaud.  

Author Response

Please see the attached PDF file. Thank you. 

Reviewer 2 Report

Authors evaluated the prevalence and prognosis of peritonsillar abscess between ESRD and non/ESRD patients. This is a large population based comparison cohort study.  However, there are several concerns.

-Introduction: In the second paragraph, authors described that PTA is different from deep neck infection. However, before and after the paragraph, there is no description about deep neck infection. Why was this paragraph inserted here? In introduction section, authors should introduce previous studies about PTA in ESRD patients, the need for this study more specifically, and difference between current and previous studies.

As it could be easily assumed that the prevalence of infectious disease could be higher in ESRD patients compared to control, specific importance of this study, and limitation of previously reported studies should be introduced.

-In Method section: There are various risk factors for PTA which were evaluated in previous studies. For examples, smoking and alcohol are important risk factors of PTA (Laryngoscope. 2020 Dec;130(12):2833-2838), and dyslipidemia also should be matched in comparison study of PTA (JAMA Otolaryngol Head Neck Surg 2019 Jun 1;145(6):530-535). These factors should be considered and matched in both cohort.

-Results and Discussion: In Results, the management for PTA is not described, and in Discussion section, authors described that both groups received aspiration or I/D. How did authors get information about treatment method? How many patients underwent aspiration and how many patients underwent I/D? Is there anyone who just underwent medical treatment only?  Is the treatment method not a contributing factor for prognosis of PTA?

If authors want to compare the prognosis of PTA as written in current manuscript, authors should provide information about exact treatment method for each patients in Result section and should consider it as one of contributing factor for prognosis of PTA in statistical analysis.

Author Response

(The authors gave the same response as above.)

Reviewer 3 Report

The article „High Risk of Peritonsillar Abscess in End-Stage Renal Disease Patients: A Nationwide Real-World Cohort Study“ focuses on the still active problems of purulent peritonsillar complications. The numbers of files are enormous, and the methods are sound.

The ideas are supported by statistics and conclusions corresponds with the methods. I a my opinion this is high quality article.

However, there are some limitation, which must be addressed.

The result are surprisingly weak. I would expected, that patients with renal disease would suffer with more complications. No differences between groups, but truly stated. The literature can be improved using some new references.

The suggestions are here.

Mejzlik, Jan & ÄŒelakovský, Petr & Tucek, L & Kotulek, M & Vrbacky, A & Matoušek, Petr & Stanikova, Lucia & Hoskova, T & Pazs, A & Mittu, P & Chrobok, Viktor. (2017). Univariate and multivariate models for the prediction of life-threatening complications in 586 cases of deep neck space infections: retrospective multi-institutional study. The Journal of Laryngology & Otology. 131. 1-6. 10.1017/S0022215117001153.

ÄŒelakovský, Petr & Kalfert, David & Tucek, Lubos & Mejzlik, Jan & Kotulek, Milos & Vrbacky, Ales & Matoušek, Petr & Stanikova, Lucia & Hoskova, Tereza & Pasz, Adam. (2014). Deep neck infections: Risk factors for mediastinal extension. European archives of oto-rhino-laryngology : official journal of the European Federation of Oto-Rhino-Laryngological Societies (EUFOS) : affiliated with the German Society for Oto-Rhino-Laryngology - Head and Neck Surgery. 271. 1679-1683. 10.1007/s00405-013-2651-5.

Author Response

(The authors gave the same response as above.)

Round 2

Reviewer 2 Report

Authors tried to response to reviewer's comments, and now, the manuscript is much improved.

This manuscript is a resubmission of an earlier submission. The following is a list of the peer review reports and author responses from that submission.